

# Impact of climate change on persistent cold-air pools in an alpine valley during the 21st century

Sara Bacer[1], Julien Beaumet[2,3], Martin Ménégoz[2], Hubert Gallée[2], Enzo Le Bouëdec[1], and Chantal Staquet[1]

[1]Univ. Grenoble Alpes, CNRS, Grenoble INP, LEGI, 38000 Grenoble, France
[2]Univ. Grenoble Alpes, CNRS, IRD, Grenoble INP, IGE, 38000 Grenoble, France
[3]Atmo Auvergne-Rhône-Alpes, 38400 Grenoble, France

**Correspondence:** Sara Bacer (sara.bacer@gmail.com)

**Abstract.** When anticyclonic conditions persist over mountainous regions in winter, cold-air pools (i.e. thermal inversion layers) develop in valleys and persist from a few days to a few weeks. During these persistent cold-air pool episodes (PCAPs) the atmosphere inside the valley is stable and vertical mixing is prevented, promoting the accumulation of pollutants close to the valley bottom and worsening air quality. It has been shown from reanalysis that the Greater Alpine Region has warmed by

three degrees over the last four decades. The purpose of this paper is to address the impact of climate change on PCAPs until the end of this century for the alpine Grenoble valleys.

The long-term projections produced with the general circulation model MPI downscaled over the Alps with the regional climate model MAR (Modèle Atmosphérique Régional) are used to perform a statistical study of PCAPs over the 21st century. The trends of the main characteristics of PCAPs, namely their duration, frequency, and intensity, are investigated for two future

scenarios, SSP2-4.5 and SSP5-8.5. We find that the intensity of PCAPs over the 21st century displays a statistically significant decreasing trend for the SSP5-8.5 scenario only, with a very weak decay rate of $0.058 \, \text{K km}^{-1} \, \text{decade}^{-1}$.

The impact of climate change on the detailed structure of PCAPs is next investigated by comparing two such episodes, in the past and around 2050 considering the worst-case scenario (SSP5-8.5). For this purpose, the WRF (Weather Research and Forecasting) model, forced by MAR, is used at a high resolution (111 m). The episodes are carefully selected so that a

meaningful comparison can be performed. We find that these episodes present similar atmospheric circulation and heat deficit across the valley depth but different atmospheric stability and (therefore) a different inversion height. The future episode is characterised by stronger atmospheric stability and a lower inversion height and about 4 degrees warmer air both close to the surface and in altitude.

Overall, this study shows that the atmosphere in the Grenoble valleys tends to be slightly less stable in the future, under the

SSP5-8.5 scenario, but that intense PCAPs can still form.

## 1 Introduction

Mountain valleys are subject to cold-air pools during wintertime when an anticyclonic synoptic regime sets in over the valley region. An anticyclonic episode is associated with mid-level warming and -often- weak synoptic winds, making the valley



atmospheric boundary layer nearly decoupled from the synoptic flow. The boundary layer dynamics is then controlled by
local thermal winds (see Whiteman and Doran, 1993, for a fuller discussion), which result from the cooling (at night) and the
warming (during daytime) of the slopes of the valley. During wintertime when solar insolation is weak and shadow effects
are important, down-slope winds due to cooling dominate over the up-slope winds, ceasing only for a few hours around noon
(f.i. Largeron and Staquet, 2016b). These down-slope winds bring cold air down to the valley bottom, thereby forming cold-
air pools which trap pollutants. These cold-air pools are persistent during anticyclonic episodes, namely they are not fully
destroyed during the day by convective motions at the ground or turbulent erosion at the top (Whiteman and McKee, 1982).
They can last from a few days to even a few weeks depending upon the length of the episode. Persistent cold-air pools are
denoted PCAPs hereafter. The link between anticyclonic wintertime episodes and PCAPs was clearly assessed by Milionis
and Davies (2008) using radiosoundings in the UK over five years and by Reeves and Stensrud (2009) using a three-year
climatology of PCAPs from valleys and basins in the Western United States. Because the temperature profile has a positive
gradient in a PCAP, namely the lapse rate is reversed with respect to the adiabatic gradient, the name *thermal inversion* is used
in the literature to refer to this profile. The qualificative ground-based, surface-based or near-surface is often added to make
the difference with elevated thermal inversions. In the literature, a PCAP is also referred to as an inversion (or a stagnation)
episode.

It has been shown from ERA5 reanalysis that the Greater Alpine Region has warmed by 3 degrees over the last forty years
(1980-2018) (Pepin et al., 2022). The impact of climate change on PCAPs is an intriguing issue. Using simple arguments,
PCAPs may indeed become more intense in a changing climate because the air will be warmer in altitude. But the PCAP
intensity may possibly hardly vary because the air close to the ground will get warmer as well. However, the latter warmer
air may also promote convective motions and mixing thereby reducing the strength of the inversion. Considering an urbanized
valley and assuming that pollutant emissions remain the same over time, each case has a very different impact on air quality.

So far, little has been published on how PCAPs can change in a future warming climate and on PCAP climatology in general.
Most studies actually focus on the occurrence of inversions, those in the past being based on observations and reanalysis.
Whiteman et al. (2014) analyse the intensity of PCAPs during winter over forty years (1971-2013) in the Salt Lake Valley
(USA) using twice-daily meteorological and air quality data; they do not find any statistically significant long-term trend. Hou
and Wu (2016) study the thermal inversions over six decades (1951-2010) simulated worldwide with reanalysis data at 2.5°
resolution and find a general increase of the occurrence of thermal inversions, except for high latitudes; such an increase is
not significant in winter. Yu et al. (2017) use the North American Regional Reanalysis (of resolution 32 km) for the period
1979-2012 to study PCAPs in valleys in the Western United States from October to March. A significant interannual variability
is found which the paper mainly aims to link to the large-scale circulation by means of statistical analysis. Rasilla et al. (2022)
use temperature measurements of two meteorological stations located at the bottom (about 600 m) and at the top (about 1900
m) of the southern Spanish Plateau to study the climatology of frequency and intensity of cold-air pools over the past sixty
years (1961-2020). No statistically significant trend is found during the night, while the intensity increases during the day
because of enhanced warming at the high-elevation site.



The impact of future climate change on thermal inversions has been addressed in very few papers, for the Po valley basin
(Caserini et al., 2017) and for southeast Australia (Ji et al., 2019). Caserini et al. (2017) analyse meteorological data recorded
at stations in the Po valley over the 1985-2013 period and outputs from a regional climate model (RCM) over the 1950-
2100 period (at resolution $0.44°$). Two different future scenarios, associated with different green-house gases emissions, are
considered: a "middle-of-the-road" one (SSP2-4.5) and an extreme one (SSP5-8.5). Caserini et al. (2017) find a weak change
in PCAPs frequency at the end of the century compared to the 1986-2005 average, of +10% for SSP2-4.5 and even less for
SSP5-8.5. Ji et al. (2019) use data from a RCM (with three sets of physical parameterizations) forced by four different general
circulation models (GCMs) from the CMIP3 database. Their objective is to analyse the impact of climate warming on inversions
at nine different locations in cities of southeast Australia (such as Canberra and Melbourne), over thirty year periods, in the
past and in the future, for a green-house gas emission scenario similar to SSP5-8.5. In the future, the results show that there
is a substantial increase in the strength of temperature inversions, implying that poor air quality episodes will be intensified.
However, the authors note that the largest differences between simulations during the second half of the century are associated
with the driving GCMs.

To the best of our knowledge, no study addresses the impact of future climate change on PCAPs forming in a mountain
valley in winter. Due to the width of these valleys, of a few tens of km in the Rocky mountains and a few kms in the Alps, an
additional downscaling is required from the scales simulated by RCMs. For instance, the Grenoble valleys considered in this
paper are about 2-3 km wide with largest width of about 6 km. The atmospheric boundary layer during PCAPs in these valleys
has been studied only for past episodes (Largeron and Staquet, 2016b, a). The impact of climate change on the atmospheric
boundary layer in a metropolis has been addressed for very few cities so far, such as London (San José et al., 2018) and Porto
(Rafael et al., 2020), for heat waves around 2050. The objective of the present paper is to address the impact of climate change
on PCAP episodes in the Grenoble valleys over the century. For this purpose, a two-fold approach, statistical and deterministic,
is followed: (i) a statistical analysis of PCAP strength and frequency over the century from RCM predictions forced by a GCM;
(ii) a detailed analysis of two PCAP episodes in the past and in the future using fine-scale numerical modelling of the valley
boundary layer forced by outputs of the RCM.

The outline of the paper is the following. Data and methodology are described in section 2. The reliability of the chain of
models to predict the stability of the Grenoble valley atmosphere is discussed in section 3. Section 4 presents the impact of
climate change on PCAPs over the 21st century, while section 5 focuses on the analysis of the two PCAP episodes. A discussion
and conclusions are reported in section 6.

## 2  Data and Methodology

### 2.1  Measurement site and meteorological data

This study focuses on the alpine area of Grenoble (France), which is the largest city in the French Alps with about half a
million inhabitants and many industries. The city is located at 210 m above sea level (a.s.l.) at the confluence of three valleys,
referred to as Grésivaudan (North-East oriented), Voreppe (North-West oriented), and Drac (North-South oriented) (Figure 1).



These valleys are 2 to 6 km wide and about 20 km (for Voreppe and Drac) and 40 km (for Grésivaudan) long. The valleys delimit three mountain chains, Belledonne, Vercors, and Chartreuse, with steep topography and high peaks up to 3000 m a.s.l. in Belledonne and 2000 m a.s.l. in Vercors and Chartreuse. In the following, the set of valleys converging to the Grenoble city will be called the *Grenoble valley system* or simply the *Grenoble valleys*, and the geographical location of the Grenoble city

will be referred to as the *Grenoble basin*.

The city of Grenoble experiences poor air quality in winter when PCAPs form and ventilation of pollutants, mainly from wood combustion and traffic, is limited. It was shown by Largeron and Staquet (2016b), when analysing the thermal inversions during nine PCAPs of the winter of 2006-2007, that these inversions display characteristics (strength and height) comparable to those of valleys in the Rocky mountains documented by Whiteman et al. (1999b).

Several ground-based automatic weather stations are installed in the Grenoble valleys, operated by the air quality agency of the région Auvergne Rhône-Alpes (Atmo AuRA) and by the French weather forecast service (Météo-France). Measurements are air temperature at 2 m from the ground ($T_{2m}$) and wind speed at 10 m, with hourly frequency. In this paper, temporal series of $T_{2m}$ recorded at two stations located at different altitudes are used to derive the vertical temperature gradient (see section 3). The two stations are Pont de Claix (PoC) at 237 m a.s.l. and Peuil de Claix (PeC) at 935 m a.s.l. (Figure 1). It was shown

in Largeron and Staquet (2016a) that the temperature field is nearly homogeneous horizontally during inversion periods, thus, a vertical gradient can be derived from stations located at different horizontal positions in the valley.

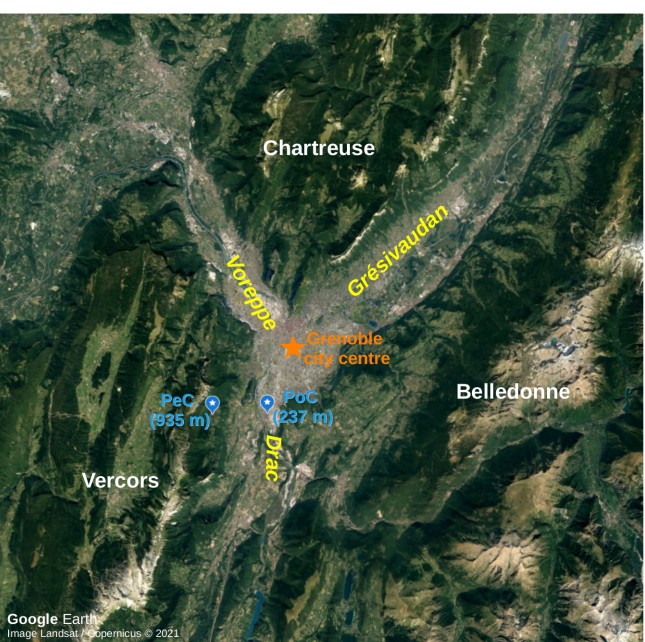

**Figure 1.** Satellite view of the Y-shape Grenoble valley system from Google Earth. The location of the weather stations is indicated in blue; the names of the valleys and mountain chains are indicated in yellow and white, respectively.



## 2.2 The model chain

In this study, three different numerical models are used. The outputs of a GCM at a resolution of $\sim 100$ km are dynamically downscaled with a RCM at a resolution of 7 km over the Alps. The latter data set is further dynamically downscaled with a third atmospheric model at a high resolution (about 100 m) to investigate the atmospheric circulation in the Grenoble boundary layer. The latter simulations are relatively short (about one week long) because of their high computing time. The details of these three atmospheric models are described below. The notation "A←B" indicates that the model A is forced by the model B, and the *model chain* is denoted as A←B←C.

### 2.2.1 The MPI-ESM1.2-HR GCM

The Max Planck Institute Earth System Model version 1.2 – Higher Resolution (MPI-ESM1.2-HR) is a coupled GCM (Müller et al., 2018). This model was run with a 100-km resolution (in the atmosphere) within the Coupled Model Intercomparison Project – Phase 6 (CMIP6, Eyring et al. (2016)), providing both historical simulations from 1850 and future projections based on different SSP-RCP scenarios (O'Neill et al., 2016). MPI-ESM1.2-HR was chosen for this study because it is one of the GCMs showing a limited bias over the Europe-North-Atlantic area (Cannon, 2020; Fernandez-Granja et al., 2021). This GCM shows also a climate sensitivity (i.e. a temperature response to a doubling of carbon dioxide emissions) compatible with the observations (Mauritsen et al., 2019), while many other CMIP6 models have instead a too large climate sensitivity (Zelinka et al., 2020). Moreover, MPI-ESM1.2-HR captures well anticyclonic episodes associated to European atmospheric blocking, as shown in Bacer et al. (2022).

Ensemble member experiments have been produced with MPI-ESM1.2-HR within the CMIP6, but due to the large computational cost of the dynamical downscaling, we consider only the first member (r1i1p1f1[1]). The experiments used in this paper are a historical run (Jungclaus et al., 2019) and two future runs based on SSP2-4.5 (Schupfner et al., 2020a) and SSP5-8.5 (Schupfner et al., 2020b). We will refer to the MPI-ESM1.2-HR model simply as "MPI" for conciseness and to the three experiments over their respective periods as MPI_HIST, MPI_SSP2 and MPI_SSP5.

### 2.2.2 The MAR RCM

The RCM used in the present study is the Modèle Atmosphérique Régional (MAR) (Gallée and Schayes, 1994; Gallée et al., 2005).

**A brief overview of the MAR model.** MAR is a hydrostatic, primitive equation model with constant sigma coordinates on the vertical axis. MAR has been developed in particular for polar regions (f.i. Gallée et al. 1996; Fettweis et al. 2017) including a multi-layer snow cover model (Brun et al. 1992) with prognostic equations for snow density, temperature, water content, and albedo that allows an estimation of the surface mass balance of ice sheets and a realistic representation of snow covered area. MAR has also been used to downscale atmospheric reanalysis in high mountain areas, over the Himalayas (Ménégoz et al.,

---

[1] where r: initial conditions; i: initialization method; p: physical scheme; and f: forcing configuration.





2013) and over the Alps (Ménégoz et al., 2020), and has been shown to provide an accurate estimation of precipitation rates including snowfall rates.

In the Alpine region, MAR is used with a 7-km resolution, which was chosen as a trade off between (i) the minimum resolution compatible with the hydrostatic approximation and the activation of the convective scheme and (ii) an accurate representation of the alpine topography and the atmospheric variables at different elevated areas. In winter, the model shows a fairly good agreement at intermediate elevations, but presents a warm bias close to the surface (of about 2.5°C) and a cold bias at higher elevations, around 3000 m a.s.l. (Beaumet et al., 2021).

**The MAR model in the present study.** The version 3.9 of MAR is used to downscale the outputs of MPI in the three experiments mentioned above, MAR←MPI_HIST (1981-2014), MAR←MPI_SSP2 (2015-2100), and MAR←MPI_SSP5 (2015-2100), for a domain extending from 1.5°E to 18.5°E and 41.5°N to 49.5°N (Figure 2, left). A spin-up of one year is required for the MAR experiments to ensure an equilibrium of the soil hydrothermal regime.

The 7-km resolution smoothes the topography, which reaches a maximum altitude of 3500 m a.s.l. over the Western Alps

(Figure 2, left), while the highest summit (Mont Blanc) peaks at 4808 m. Regarding the representation of the Grenoble valley system, the surrounding summits reach at most 1700 m a.s.l. in Belledonne and 1200 m a.s.l. in Vercors and Chartreuse. Nevertheless, this resolution allows for a good representation of the Y-shape of the Grenoble valley system (Figure 2, right). As for Grenoble basin, the lowest altitude is 422 m a.s.l., i.e. about 200 m higher than the real altitude. The grid box in the center of the basin, representative for the Grenoble city centre, is labeled with "GR" in Figure 2 (right).

The vertical resolution is distributed through 24 levels, from the surface to 0.1 hPa. The outputs of the three experiments MAR←MPI_HIST, MAR←MPI_SSP2, and MAR←MPI_SSP5 are available with daily resolution at different elevations (2, 10, 50, 100 m a.s.l.) and pressure levels (925, 850, 800, 700, 600, 500, 200 hPa). A data set for a limited number of variables and levels is available online (see *Data availability* at the end of this paper).

These daily outputs are used to identify thermal inversions along the 21st century in section 2.3.2 and to analyse the trends

of inversion characteristics in section 4. Experiments MAR←MPI_HIST and MAR←MPI_SSP5 are used to laterally force the third atmospheric model applied at high resolution; in this case, MAR is rerun saving hourly data distributed on 36 levels for the period of the selected PCAPs (more details on the MAR downscaling are provided in the supplementary material).

### 2.2.3   The WRF model

The Weather Research and Forecasting (WRF) model is a state-of-the-art atmospheric modeling system designed for both mete-

orological research and numerical weather prediction. It is a fully compressible, non-hydrostatic model with a terrain-following, hybrid sigma-pressure vertical coordinate and the Arakawa-C grid staggering. The dynamical solver is the Advanced Research WRF (ARW). A detailed description of the model can be found in Skamarock et al. (2019). In the present work, version 4.1 is used, with the 3rd-order Runge-Kutta time integration scheme and a 5th-order Weighted Essentially Non-Oscillatory (WENO) scheme with a positive definite filter for the advection terms. The model topography is based on the NASA Shuttle Radar

Topography Mission (SRTM) digital elevation (version 4) with a resolution of 3 arc seconds (i.e. approx. 90 m).



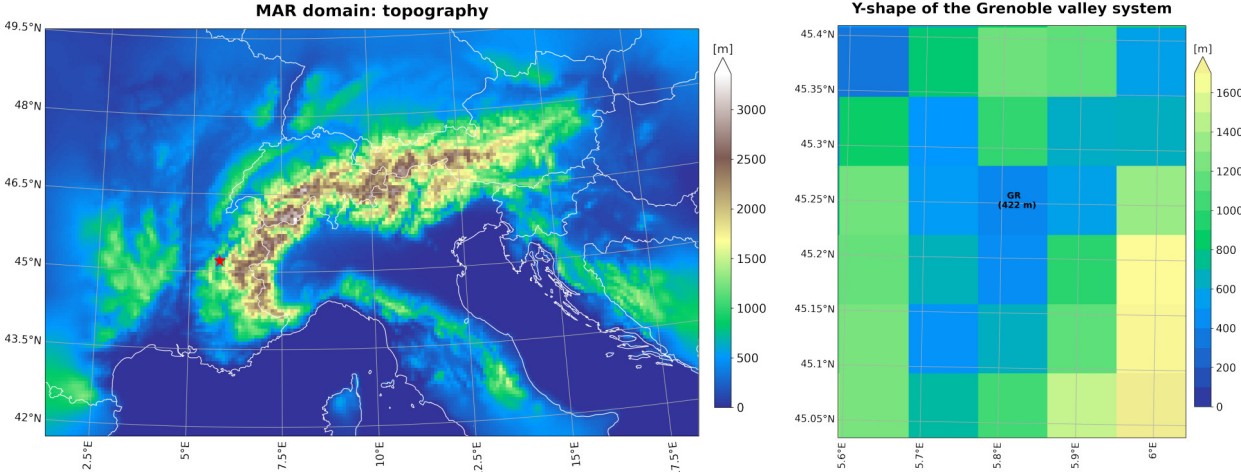

**Figure 2.** MAR domain with horizontal resolution of 7 km x 7 km (left); the red star indicates the location of Grenoble. Representation of the Y-shape Grenoble valley system in MAR (right); the grid box labeled with GR is in the center of the Grenoble basin and stands at the location of the Grenoble city, with altitude 422 m a.s.l.

Sub-kilometer simulations with WRF have been performed for worldwide mountain valleys, f.i. the Salt Lake Valley at 250 m of resolution in the USA (Crosman and Horel, 2017). In Europe, alpine valleys such as the Bolzano basin (300 m, Tomasi et al. (2019)), the Passy valley (111 m, Arduini et al. (2020); Quimbayo-Duarte et al. (2021)), and the Inn valley (40 m, Umek et al. (2021)) have been addressed. In this study, thanks to the definition of three online one-way nested domains (Figure 3, left), a resolution of 111 m is reached in the innermost domain covering the Grenoble valleys (Figure 3, right). Table 1 provides the details of spatial and temporal resolutions in each domain. Along the vertical, 91 model levels are defined up to 50 hPa (approx. 19 km a.s.l.). The thickness of the vertical model layers ($\Delta z$) increases approximately linearly in the first 50 layers before being stretched; the first 1000 m above ground level (a.g.l) are discretized into 22 model levels, with the first mass point being located at about 17 m above the surface.

The presence of steep slopes can generate numerical instabilities, even for high vertical resolution (see f.i. Connolly et al., 2021, for a discussion). To prevent these instabilities, a smoothing algorithm is applied to the topography of the innermost domain based on an optimization method in which the maximum slope angle is prescribed (Le Bouëdec, 2021). The latter angle is set to $28°$ in the present work. For that angle, the smoothing procedure affects mainly the highest peaks, in the Belledonne chain in particular; the lowest parts of the valley slopes and the valley floors are only moderately or not affected. As discussed in Le Bouëdec et al. (2022), quite remarkably, the atmospheric circulation inside the valley is found similar to the case of a maximum slope angle of $42°$.

Since an accurate land use description is also important to correctly resolve the valley winds (Schmidli et al., 2018), the updated and high resolution Corine Land Cover (CLC2018) data set with a horizontal resolution of 100 m is used. Land surface processes are modelled using the Noah land surface model (Chen and Dudhia 2001), with four soil layers, and the



Monin-Obukhov similarity theory (Jiménez et al., 2012) is used to couple the land surface to the atmosphere. Slope effects on radiation and the topographic shading are also considered.

For the two outermost domains the planetary boundary layer (PBL) is parameterised with the Yonsei University (YSU) scheme (Hong et al., 2006). For the higher-resolution simulations in the innermost domain, a three-dimensional turbulent kinetic energy (TKE) 1.5-order closure scheme is used (and no PBL scheme).

The WRF←MAR←MPI model chain has been designed to simulate PCAP episodes during past and future climate (the dynamical downscaling of MAR with WRF is described in the supplementary material). In particular, two WRF simulations have been run to describe one PCAP in the past and one around the middle of the 21st century. The identification and the selection of the episodes is described in subsection 2.3.2. The WRF simulations are five days long, after a spin-up time of one day.

| Set-up | Description | D1 | D2 | D3 |
|--------|-------------|-----|-----|-----|
| $\Delta t$ [s] | model integration time step | 12 | 2 | 0.2 |
| $\Delta x$ [m] | horizontal resolution | 3000 | 1000 | 111.111 |
| $\Delta z_{m0}$ [m] | minimum and maximum height of the first mass point | [16.0-17.9] | [15.6;17.9] | [16.5;17.8] |
| $n_z$ | number of vertical levels | 91 | 91 | 91 |
| $n_x = n_y$ | number of grid points in the W-E and S-N directions | 208 | 340 | 406 |

**Table 1.** Numerical parameters of the WRF simulations.

## 2.3 Detection of persistent inversions

### 2.3.1 Identification of PCAPs from MAR←MPI over the 21st century

In order to detect PCAP episodes in the Grenoble valley system during winter, we rely on the methodology followed by Largeron and Staquet (2016a). These authors study persistent inversions in the Grenoble valley boundary layer during the winter of 2006-2007. Winter encompasses the months from November to March and is denoted as NDJFM in their paper, and we adopt here the same definition and notation. Largeron and Staquet (2016a) use hourly temperature measurements recorded by meteorological stations located in the Grenoble valleys at different elevations and define PCAPs as periods during which the 24-h running mean of the vertical temperature gradient ($\Delta T/\Delta z$) is above the winter average, close to $-3$ K km$^{-1}$, for at least three days. By assuming horizontal homogeneity of temperature within the valleys, $\Delta T/\Delta z$ is defined as the ratio of the $T_{2\mathrm{m}}$ difference between one station at high elevation and one station at low elevation and the difference in altitude of the two stations. Such a ratio is shown to be a good indicator of the stability of the atmosphere when an inversion occurs (Largeron and Staquet, 2016a).

Since long-term MAR outputs are daily means and temperature at high elevation is available only at specific pressure levels (this is common in RCMs and GCMs), we adapt the method described above to our data set. We observe that the mean heights of the 925 hPa and 850 hPa levels for the past thirty years (1981-2010) are equal to 787±91 m a.s.l. and 1472±95 m a.s.l.,





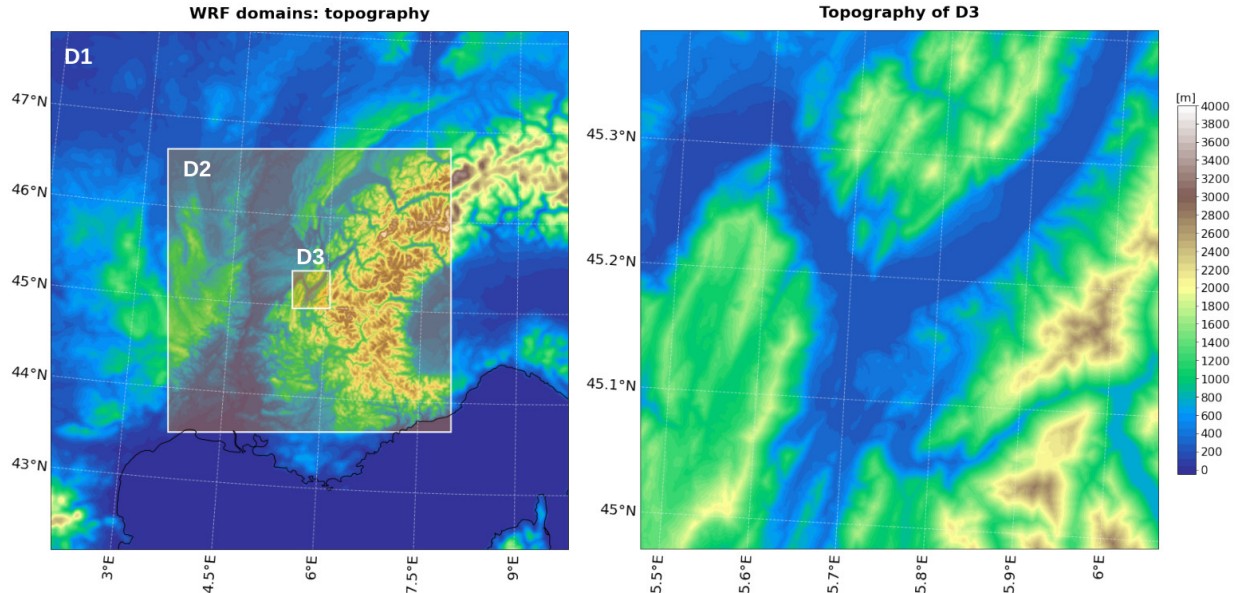

**Figure 3.** Topography in the three nested WRF domains (left) and in the innermost domain, D3, with horizontal resolution of 111 m x 111 m (right).

respectively. The 850 hPa level is on average higher than Vercors and Chartreuse in MAR (Figure 2, right), while an inversion top is usually close to the mean height of the mountains surrounding the valley (Whiteman, 1982; Largeron and Staquet, 2016b; Rasilla et al., 2022). On the contrary, the 925 hPa level should always be below the height of the thermal inversions observed in Grenoble. Since the temperature profile is approximately linear during inversions (Whiteman, 1982), this level is chosen to compute the vertical temperature gradients modeled by MAR, which we assume to be representative for the entire inversion
depth. Therefore, the modeled $\Delta T/\Delta z$ is defined as:

$$(\Delta T/\Delta z)_{\mathrm{MAR}} = \frac{T_{925\mathrm{hPa}} - T_{2\mathrm{m}}}{Z_{925\mathrm{hPa}} - z_{2\mathrm{m}}} \tag{1}$$

where $T_{925\mathrm{hPa}}$ is the temperature at 925 hPa, $Z_{925\mathrm{hPa}}$ is the geopotential height at 925 hPa and $z_{2\mathrm{m}}$ is the altitude at 2 m above the ground. $(\Delta T/\Delta z)_{\mathrm{MAR}}$ is computed at the GR grid box; hence $z_{2\mathrm{m}} = 424$ m a.s.l.

    PCAP episodes are identified with the following criterion:

$(\Delta T/\Delta z)_{\mathrm{MAR}} > \langle(\Delta T/\Delta z)_{\mathrm{MAR}}\rangle_{30\mathrm{winters}}^{\mathrm{HIST}}$ for at least 5 consecutive days $\tag{2}$

where $\langle(\Delta T/\Delta z)_{\mathrm{MAR}}\rangle_{30\mathrm{winters}}^{\mathrm{HIST}}$ is the winter average computed over 30 years (1981-2014) with MAR←MPI_HIST, rounded to $-3$ K km$^{-1}$. This value is fully consistent with the winter average computed from the observations over 30 years (1985-2014) (see section 3). Also, it is in agreement with Largeron and Staquet (2016a), who considered only one winter of observations, and close to Le Bouëdec (2021) ($-2.5$ K km$^{-1}$), who considered six winters of observations. Note that in the two latter
references, the highest elevation station to compute the gradient is at a higher altitude than PeC, about 1700 m.





A similar method is also used by Iacobellis et al. (2009) to detect temperature inversions in California. These authors consider the temperature at 850 hPa and detect inversions if a ratio similar to (1) is greater than zero. By applying the same relation and condition for the Po Valley basin, (Caserini et al., 2017) obtain a strong underestimation of inversion frequency with the regional climate model they consider. Largeron and Staquet, 2016a, who tested different thresholds, also find that the condition
"greater than zero" is too restrictive as too few PCAP episodes are detected. This discussion stresses the importance of testing and choosing the best threshold for the inversion detection.

In this work, PCAPs are identified for the entire 21st century, from 1981 until 2100, using MAR←MPI_HIST, MAR←MPI_SSP2 and MAR←MPI_SSP5.

### 2.3.2 Selection of two PCAP episodes to be simulated with WRF in the past and around 2050

The careful selection of two PCAPs, one in the past and one around 2050, to be simulated with WRF is an essential step to allow a meaningful comparison of the two episodes (see section 5). Therefore, criteria are defined here in order to select two PCAPs that present common characteristics.

First of all, the episodes must present a similar synoptic situation. The sensitivity of inversion activity to large-scale atmospheric circulation and the higher occurrence of ground-based temperature inversions during an anticyclonic regime have indeed already been demonstrated, as stressed in the Introduction. Wintertime anticyclonic conditions over the Grenoble area
can occur during the Scandinavian atmospheric blocking, when it expands southwards. In this work, atmospheric blocking episodes are identified by applying the so-called weather type decomposition (WTD), a methodology that classifies the atmospheric circulation into discrete weather regimes (Michelangeli et al., 1995). The WTD consists of two steps: the dimensional reduction of the data set via the principal component analysis (PCA) and the clustering via the $k$-means algorithm. The data
set involves here daily anomalies of geopotential height at 500 hPa. The number of clusters ($k$) is set to 4, to detect the four well-known weather types of the Euro-Atlantic sector (positive and negative North Atlantic Oscillations, Atlantic ridge, and Scandinavian atmospheric blocking).

In the present work, the WTD and the identification of blocking episodes follow the methodology used in Bacer et al. (2022). We apply the WTD to the data sets produced by MPI_HIST over the period 1981–2014 and by MPI_SSP5 over two successive
30-year periods, 2015–2045 and 2035–2065. We next select the blocking episodes that are at least 10 days long and, among them, retain those satisfying the following criteria:

1. Since these episodes are centered over western or northern Europe, in order to consider anticyclonic conditions in Grenoble, only those days characterised by daily sea level pressure (SLP) higher than 1030 hPa in Grenoble are kept; this criterion is applied by using the temporal series of SLP extracted at the GR grid box of the MAR outputs.

2. Only those days with $(\Delta T/\Delta z)_{\mathrm{MAR}} > -3\,\mathrm{K\,km^{-1}}$ in GR are considered.

3. Conditions 1 and 2 must be satisfied for at least 5 consecutive days to define a PCAP.



4. The PCAPs must occur during the same period of the year, so that the incident solar radiation, which drives the thermal forcings of the valley circulation, is comparable among the episodes.

In this work, we look for PCAPs preferably around the years 2000 and 2050, in order to consider one episode in the near past
and one episode in the middle of the century. We find that only two episodes satisfy all criteria, one in 1988 (14–21/12/1988) and one in 2043 (4–15/12/2043), as visible in Figure S1. We decide to consider 5 days in the center of the episodes for the WRF simulations, therefore, the dates of the two selected PCAP episodes are: 16–20 December 1988 and 8–12 December 2043.

During both episodes, the daily blocking patterns show a strong positive geopotential anomaly centered over central Europe, extending to the Iberian peninsula (Figure S2). The winds over South-East France are lower than $10 \text{ m s}^{-1}$ except during the
first two days of the episode in 1988, when the winds are stronger. In particular, the atmospheric blocking is not well structured in the first day (16/12/1988) yet, and westerlies still cross France zonally; this will affect the structure of the thermal inversions formed in the Grenoble valleys during these days (see section 5).

## 3 Reliability of MAR←MPI to predict the stability of the valley atmosphere

Before analysing the inversions predicted by MAR in the future and downscaling MAR with WRF, an essential step is verifying
the reliability of MAR←MPI in the past. For this purpose, the daily temperature gradient computed with MAR ←MPI_HIST (see definition (1)) is compared with the observed vertical temperature gradient, noted $(\Delta T/\Delta z)_\text{obs}$, during NDJFM over the period 1985–2014. This gradient is computed with daily means of $T_\text{2m}$ measured at the two weather stations presented in subsection 2.1 as

$$(\Delta T/\Delta z)_\text{obs} = (T_\text{2m}^\text{PeC} - T_\text{2m}^\text{PoC})/(z^\text{PeC} - z^\text{PoC}), \tag{3}$$

$z$ being the altitude of the two stations (237 m a.s.l. and 935 m a.s.l.).

Figure 4 displays the probability density functions (PDFs) of both data sets over the 30 winters of the 1985–2014 period, which contain 4447 days. The agreement between the PDFs is remarkable. The mean values are equal to $-3.2 \text{ K km}^{-1}$ for the model results and to $-3.1 \text{ K km}^{-1}$ for the observations. The winter days characterised by $\Delta T/\Delta z > -3 \text{ K km}^{-1}$ represent 43.5% of the data set according to MAR and 41.6% according to the observations, which corresponds to a difference of about
80 days. The MAR distribution is slightly shifted to the right with respect to the observations until about $4 \text{ K km}^{-1}$ but presents a shorter tail for higher values (the number of winter days with $\Delta T/\Delta z > 4 \text{ K km}^{-1}$ represents 2.2% of the MAR data and 5.7% of the observation). This underestimation of the model could be due to the warmer bias of MAR close to the ground (see subsection 2.2.2), which leads MAR to slightly underestimate the atmospheric stability in the valley. Moreover, it should be reminded that the level of the valley bottom in MAR is about 200 m higher than the actual topography (see subsection 2.2.2);
during inversion days (when temperature increases with altitude), this could also cause an overestimation of the modeled $T_\text{2m}$, contributing to reduce $(\Delta T/\Delta z)_\text{MAR}$.

A further comparison between model results and observations is performed by considering only the PCAP episodes (defined by criterion (2)) during the same 30 winters. The total number of days belonging to PCAPs derived from MAR is 886, while the



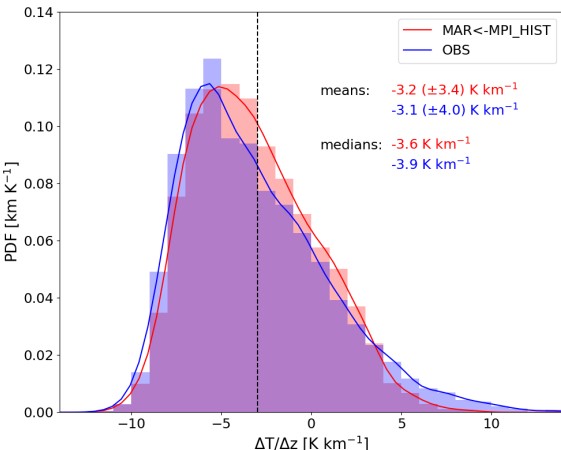

**Figure 4.** Normalized probability density functions (PDFs) of daily temperature gradient computed with MAR←MPI_HIST, *i.e.* $(\Delta T/\Delta z)_{\mathrm{MAR}}$ defined by (1) (red), and observations, *i.e.* $(\Delta T/\Delta z)_{\mathrm{obs}}$ defined by (3) (blue), over 30 years (1985-2014) during ND-JFM. The vertical line indicates the value $-3\,\mathrm{K\ km}^{-1}$ (i.e. the threshold used to identify the PCAP episodes); the standard deviation is in parenthesis.

one derived from the observations is 1040. Since the number of winter days with $\Delta T/\Delta z > -3\,\mathrm{K\ km}^{-1}$ is similar as indicated

above, we deduce that several of these days in MAR are not consecutive and, therefore, do not belong to a PCAP episode. The total number of episodes, 119 according to MAR and 91 according to the observations, indeed indicates that MAR simulates shorter episodes. This is confirmed by the PDF of the duration of the PCAPs in Figure 5 (left). Since this analysis concerns the right tails of the PDFs in Figure 4, only the median values are computed. The simulated and observed PCAPs are finally compared in terms of *intensity*, defined as the temporal mean of the daily vertical temperature gradients over the duration of

the PCAP. As expected from Figure 4, the modeled PCAPs tend to be less intense than the observed ones, with an intensity median smaller by $0.2\,\mathrm{K\ km}^{-1}$ (Figure 5, right).

     Overall, we can conclude that MAR←MPI_HIST reproduces well the vertical temperature gradient in the Grenoble basin. Therefore, MAR←MPI can be reasonably used to (i) analyse future long-term thermal inversions and (ii) force WRF. Moreover, we prove that definition (1), with temperature at 925 hPa, is a good alternative to compute the vertical temperature gradient in

the absence of temperature at specific altitudes or vertical temperature profiles.

## 4   Impact of climate change on inversions in the Grenoble basin over the 21st century

### 4.1   Trends of $\Delta T/\Delta z$ over the 21st century

We analyse the impact of climate change on the vertical temperature gradient in the centre of the Grenoble valleys (i.e. in GR) over the winters during the 21st century. We simplify the notation by referring to the 120-year long simulations from 1981 to





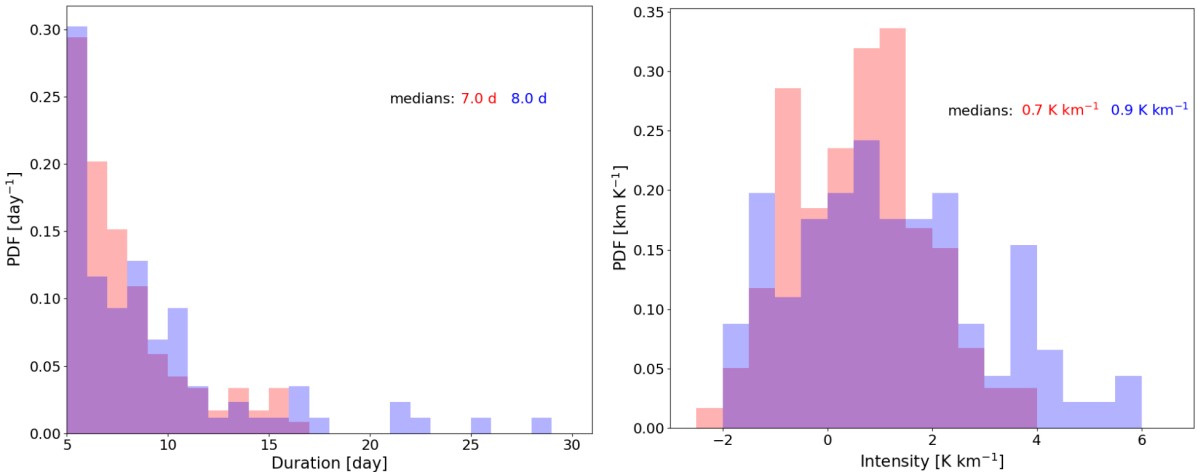

**Figure 5.** PDF of duration (left) and intensity (right) of the PCAP episodes identified over 30 years (1985-2014) during NDJFM with MAR←MAR_HIST (red) and observations (blue).

2100 as MAR←MPI_SSP2 and MAR←MPI_SSP5 (thus abandoning MAR←MPI_HIST as this historical run is common to both periods).

The temporal series of $(\Delta T / \Delta z)_{\mathrm{MAR}}$ during this period are presented in Figure 6. Under both future scenarios, the series have a temporal mean of about $-3$ K km$^{-1}$, with a standard deviation also equal to about $-3$ K km$^{-1}$. They both present a statistically significant negative trend indicating a decreasing atmospheric stability, with the slope of the trend being larger for MAR←MPI_SSP5 than for MAR←MPI_SSP2 ($-0.059 \pm 0.007$ K km$^{-1}$ per decade versus $-0.017 \pm 0.007$ K km$^{-1}$ per decade), consistent with global warming being larger for SSP5-8.5 than for SSP2-4.5. The analysis of $(\Delta T / \Delta z)_{\mathrm{MAR}}$ distributions over 30-years around 2000, 2050, and 2085 highlights a gradual decrease of strong inversion days in the worst-case scenario (Figure S3, right).

Trends of the $(\Delta T / \Delta z)_{\mathrm{MAR}}$ temporal series obtained with MAR←MPI_SSP2 and MAR←MPI_SSP5 are also computed in sliding windows with a 10-year step and of variable length, from 30 years to the length of the entire time series (Figure 7, first row). In the past, three windows can be considered: 1981-2010 and 1991-2020 (30-year long) and 1981-2020 (40-year long). For these periods, the vertical temperature gradient trends are negative, although not always statistically significant. Figure 7 (first row) confirms the long-term tendency of the vertical temperature gradient to decrease, thereby reducing the atmospheric stability in the valley, especially for SSP5-8.5. Under the latter scenario indeed, the trends are almost always negative whatever the window and always statistically significant with windows longer than 70 years. The slope coefficients are larger in the second half of the century, with values up to $-0.208 \pm 0.036$ K km$^{-1}$ per decade with windows of 40 years. With respect to this scenario, the trends in MAR←MPI_SSP2 are less often statistically significant; the slope coefficients are smaller and always negative with windows longer than 40 years.



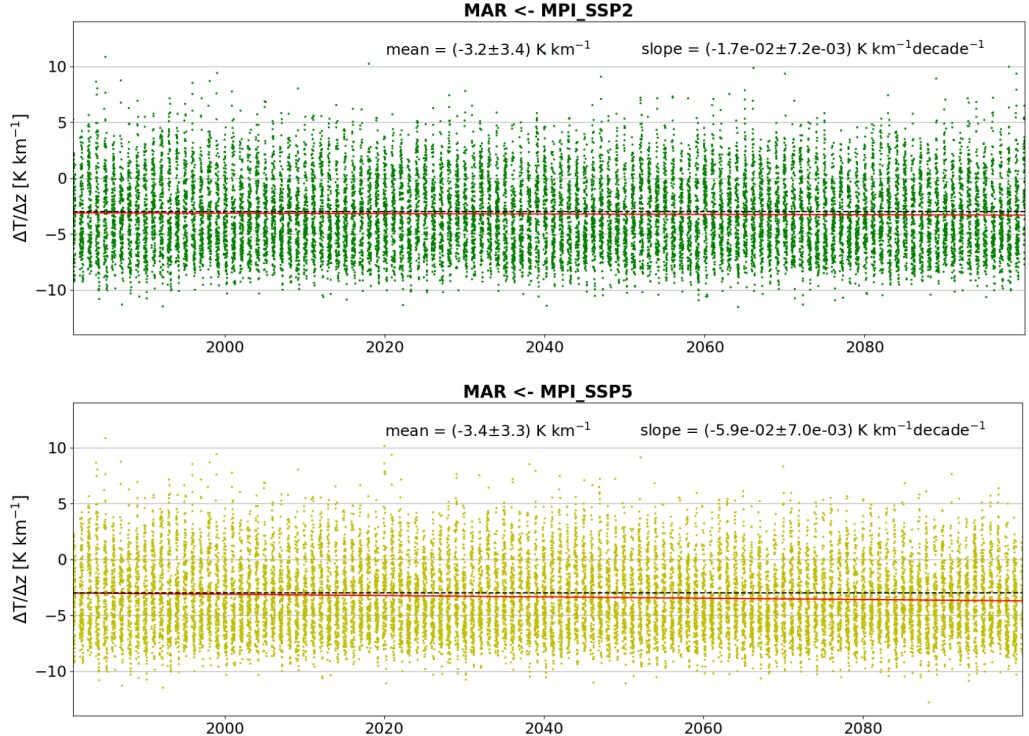

**Figure 6.** Temporal series of daily winter $(\Delta T/\Delta z)_{\mathrm{MAR}}$ for MAR←MPI_SSP2 (top) and MAR←MPI_SSP5 (bottom). The horizontal black dashed line refers to the value $\Delta T/\Delta z = -3\,\mathrm{K\,km^{-1}}$. The red line is the trend (which is statistically significant at 95%, with p-value equal to 0.017, for SSP2-4.5 and at 99%, with p-value of the order of $10^{-17}$, for SSP5-8.5). The mean and the slope of the trend ($\pm$ standard deviation) refer to the entire time series (i.e. 120 winters).

In order to study the behaviour of $(\Delta T/\Delta z)_{\mathrm{MAR}}$, the temporal series of $T_{2\mathrm{m}}$ are displayed in Figure S4. As expected, the
near-surface air temperature is projected to increase, especially for the worst-case scenario. The trends computed for $T_{2\mathrm{m}}$ in the same sliding windows as for $(\Delta T/\Delta z)_{\mathrm{MAR}}$ are displayed in Figure 7 (second row). Interestingly, this Figure shows that $T_{2\mathrm{m}}$ increases until the middle of the century under the SSP2-4.5 scenario, while the increasing trend persists until the end of the century under SSP5-8.5. Regarding the temperature at $Z_{925\mathrm{hPa}}$, we could not perform a similar trend analysis as the height of this pressure level is variable in time. However, since the vertical temperature gradient considered in this work concerns the
lowest part of the troposphere (below $\sim 800$ m a.s.l., see subsection 2.3.1) and since the ground-based inversions, governed by radiative cooling, are particularly sensitive to surface temperature, it is reasonable to suppose that the future tendency to a decreasing atmospheric stability in the valley is attributable to the increasing surface temperature (Bailey et al., 2011).





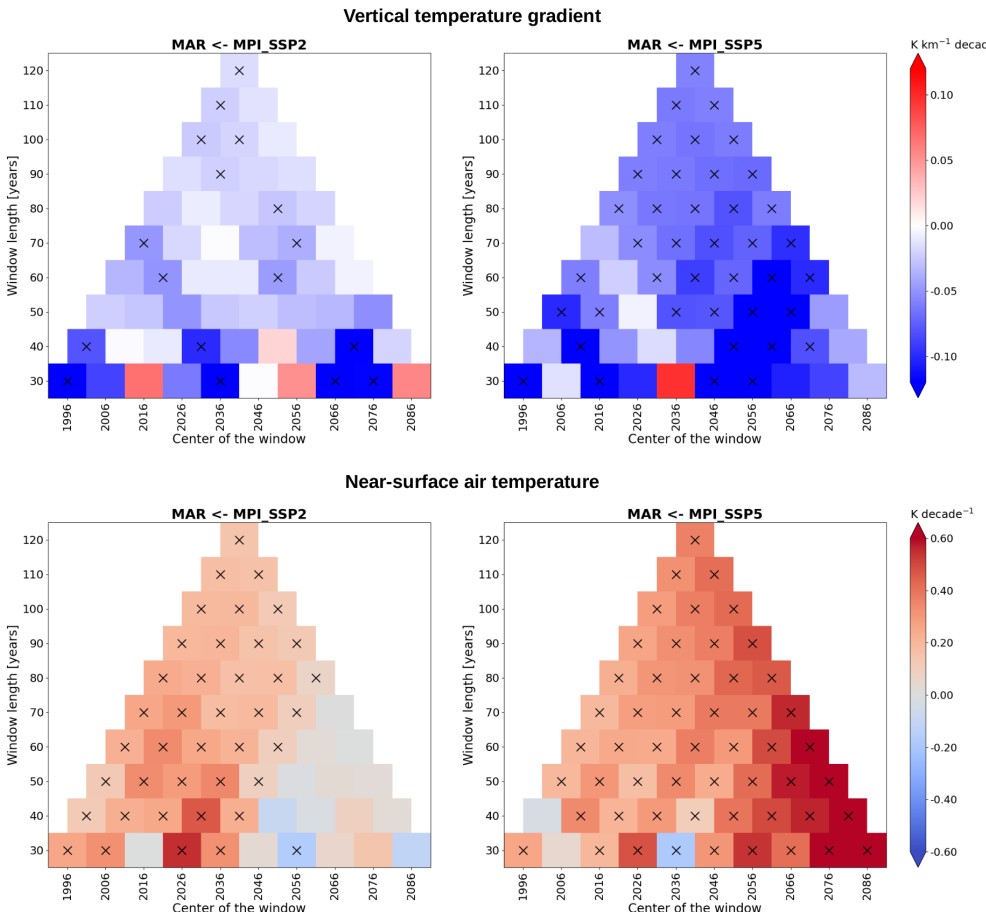

**Figure 7.** Slope coefficients of the trends computed in sliding windows of variable length for $(\Delta T / \Delta z)_{\mathrm{MAR}}$ (first row) and $T_{\mathrm{2m}}$ (second row) for MAR←MPI_SSP2 and MAR←MPI_SSP5. The windows have a minimal length of 30 years; they slide with a step of 10 years along the entire temporal series (1981-2100). The windows have also a variable length, increasing by 10 years until they reach the maximum length equal to the length of the entire temporal series. The years that are central in the windows are on the x-axis, the length of the windows is on the y-axis. The black crosses mark trends that are statistically significant at 95%.

## 4.2 Characteristics of PCAP episodes over the 21st century

We now investigate the impact of climate change on the characteristics of the PCAP episodes identified over the 21st century
(see criterion (2)). We focus on the intensity (i.e. the temporal mean of the daily vertical temperature gradients over the duration of the PCAP), the duration and the frequency (i.e. the number of episodes in a given time period). Figure 8 displays the temporal series of these quantities per year, i.e. the annual mean intensities, mean durations, and frequencies, where "annual" always refers to wintertime. The time series of intensity and duration are displayed in Figure S5 for all identified PCAPs. All time series show a large interannual variability (as in Yu et al. (2017)). As expected from the previous analysis, PCAPs will be less



intense for the worst-case scenario, changing from a 30-year mean of $0.62 \pm 1.26$ K km$^{-1}$ around the year 2000 to a 30-year mean of $0.27 \pm 1.06$ K km$^{-1}$ at the end of the century (Table 2). Also in this case, the decrease is more evident in the second half of the century (Table 2 and Figure S5). On the contrary, PCAP intensities for the SSP2-4.5 scenario do not present any significant change, and extreme episodes occur all along the century (Table 2 and Figure S5).

The annual mean PCAP duration (Figure 8, middle row) varies between 5 and 10 days during the historical period, while it can be longer in the future for both scenarios (up to 15 days under SSP2 and 17 days under SSP5), with no statistically significant trend. Considering all PCAPs occurring in periods of 30 years, the mean duration is between 7 and 8 days for both scenarios (Table 2).

The annual number of PCAPs (Figure 8, bottom row) varies between 2 and 6 during the historical period and is comprised (almost always) between 1 and 7 during both future periods. More precisely, this quantity shows a statistically significant negative trend for the worst-case scenario only. This is evident also in Table 2, where the total number of episodes over a 30-year period decreases from 121 in the past to 94 at the end of the century. Since the PCAP duration remains essentially stable, this reduction implies that the annual number of inversion days also tends to decrease.

| Quantity | 30-yr period | SSP2 | SSP5 |
|---|---|---|---|
| Mean intensity [K km$^{-1}$] | around 2000 | \multicolumn{2}{c}{$0.62 \pm 1.26\,[-2.22; 3.60]$} |
| | around 2050 | $0.37 \pm 1.13\,[-2.18; 3.21]$ | $0.51 \pm 1.16\,[-2.38; 3.43]$ |
| | around 2085 | $0.49 \pm 1.22\,[-2.02; 4.47]$ | $0.27 \pm 1.06\,[-2.01; 2.51]$ |
| Mean duration [days] | around 2000 | \multicolumn{2}{c}{$7.4 \pm 2.8\,[5; 16]$} |
| | around 2050 | $7.6 \pm 3.3\,[5; 24]$ | $7.9 \pm 3.7\,[5; 29]$ |
| | around 2085 | $8.0 \pm 4.1\,[5; 27]$ | $7.3 \pm 2.7\,[5; 18]$ |
| Number of episodes | around 2000 | \multicolumn{2}{c}{121} |
| | around 2050 | 122 | 103 |
| | around 2085 | 109 | 94 |

**Table 2.** Mean values ($\pm$ standard deviation and $[\mathrm{minimum; maximum}]$) of intensity, duration and number of the PCAP episodes occuring over 30-year periods (see also Figure S5). The terms "around 2000" refer to the 1985–2014 period, "around 2050" to the 2035–2064 period, and "around 2085" to the 2070-2099 period.

## 5 Vertical structure of two persistent inversion episodes in the Grenoble valleys

The analysis now focuses on five days of two PCAP episodes that have been simulated at high resolution with the model chain WRF←MAR←MPI_SSP5 (see subsection 2.2.3 for the WRF setup). These five days are in the middle of each PCAP episode, when the PCAP intensity is largest. One episode occurs in the past (16-20 December 1988) and the other one in the future (8-12 December 2043) (see subsection 2.3.2 for the PCAP selection criteria). In the following, these five-day periods are referred to as Ep1988 and Ep2043, respectively.





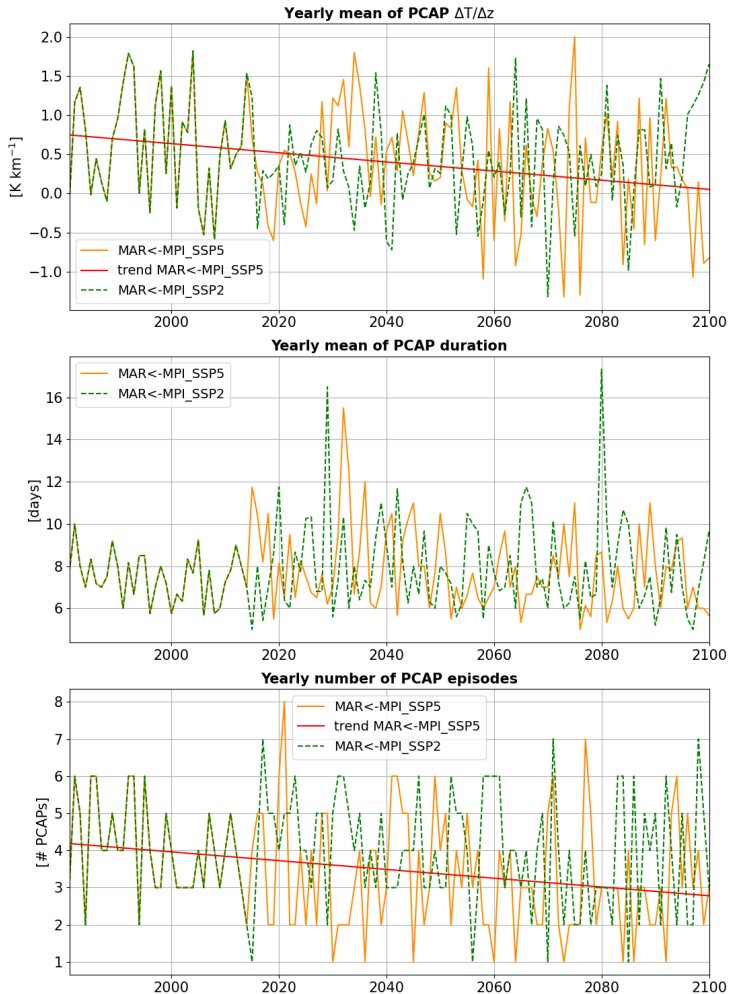

**Figure 8.** Temporal series of annual means of PCAP intensities (top), annual means of PCAP durations (middle), and annual number of PCAP episodes (bottom). The trends in red are statistically significant at 99%, with p-value equal to 0.002 and slope $(-0.058 \pm 0.019)\mathrm{K\ km^{-1}decade^{-1}}$ (top) and p-value equal to 0.003 and $(-0.118 \pm 0.039)\#\mathrm{PCAPs\ decade^{-1}}$ (bottom).

## 5.1  General features of the inversions

The general features of the inversions are displayed in Figure 9 through vertical profiles (up to 2000 m a.g.l.) of temperature and horizontal wind speed over the five selected days of each episode. These fields are averaged over the two main valleys of the Grenoble valley system, Grésivaudan and Voreppe, and over the Grenoble basin (these areas are displayed in Figure 10). The averaged profiles thus obtained are assumed to be representative of each valley section.

The arrival of the anticyclone over the Grenoble basin is associated with that of a warm air mass around 1000 m a.g.l., from 370  December 18 in Ep1988 and from December 8 in Ep2043. This warm air layer persists at mid-altitude over the remaining days





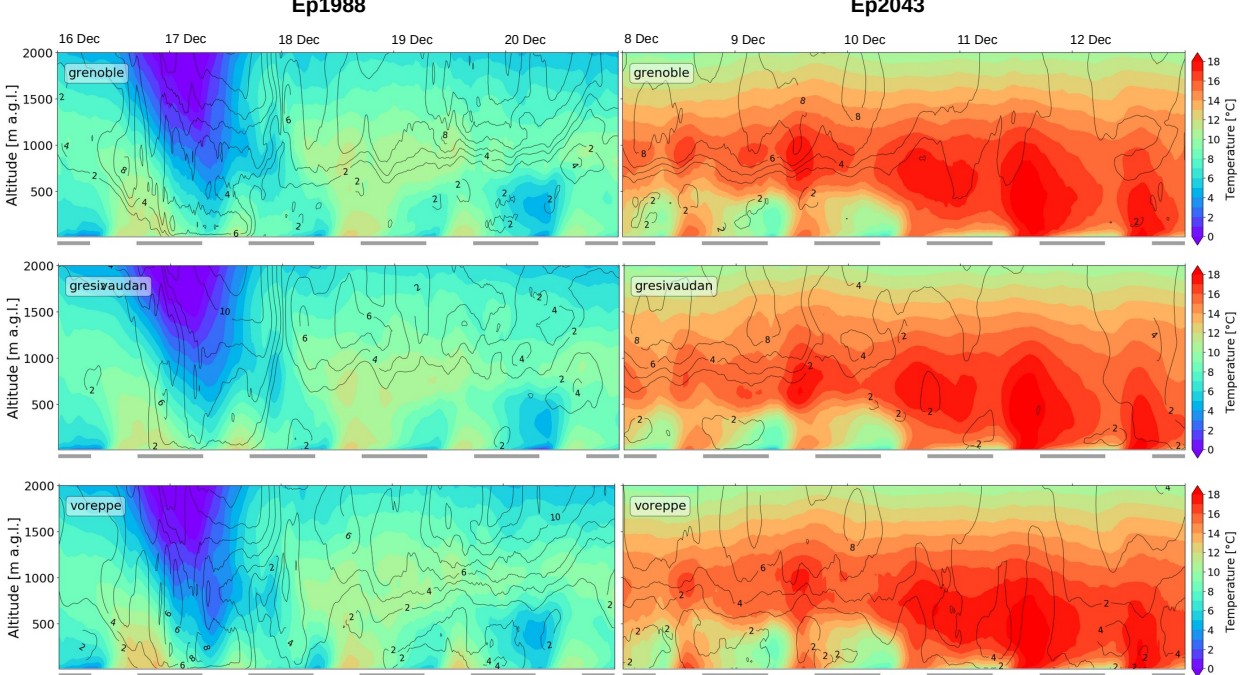

**Figure 9.** Vertical profiles of temperature (color shading) and wind speed (black contours from 0 to $14\,\mathrm{m\,s^{-1}}$, every $2\,\mathrm{m\,s^{-1}}$) as functions of time for Ep1988 and Ep2043. These quantities are spatially averaged over the Grenoble basin, Grésivaudan valley and Voreppe valley (these areas are displayed in Figure 10). The horizontal grey lines along the x-axis indicate the nighttime period, between 17:00 UTC and 7:00 UTC.

of the episode. The striking feature of Figure 9 is the difference in temperature between the two episodes. Around 800-1000 m, the temperature reaches $\simeq 17^\circ$C in Ep2043 and $\simeq 13^\circ$C in Ep1988. Close to the surface, where a diurnal cycle sets in, a similar temperature difference to mid-altitude is observed, namely $T_{2\mathrm{m}}$ is about 4 degrees higher in Ep2043 than in Ep1988. This is consistent with the $T_{2\mathrm{m}}$ projection for SSP5-8.5 (see Figure S4, bottom). In overall then, Ep2043 is much warmer than Ep1988

at all altitudes displayed in Figure 9.

The along-valley wind is very weak in the first 700 m a.g.l. for both episodes, less than $2\,\mathrm{m\,s^{-1}}$. It increases above this height for the two episodes, still remaining lower than $\simeq 8\,\mathrm{m\,s^{-1}}$, consistent with the synoptic regime being anticyclonic. Hence, as opposed to the temperature fields, the atmospheric circulation in the valley system is similar in both episodes (see also Figure S6).

Although the five days of the PCAP episodes were carefully chosen (see the criteria discussed in subsection 2.3.2) and were in the middle of an anticyclonic period, some wind intrusion occurs during Ep1988: stronger winds and cold air enter in the valleys on 17 December 1988, perturbing the valley atmosphere. Note that this event could not be detected from the large-scale wind pattern at 500 hPa (Figure S2).



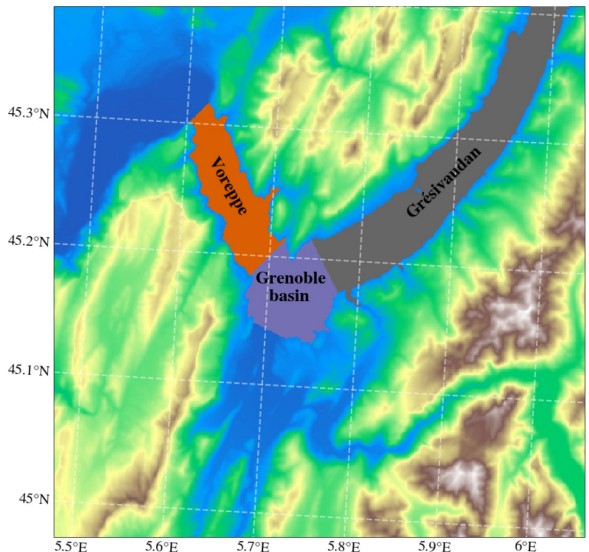

**Figure 10.** Areas considered for the spatially-averaged fields in section 5.

## 5.2 Height and strength of the inversions

The three main characteristics of the PCAPs, atmospheric stability, inversion strength, and inversion height, are now considered
for a more quantitative analysis.

   The atmospheric stability is expressed in terms of the vertical gradient of potential temperature, $\partial\theta/\partial z$, with respect to the
adiabatic lapse rate for dry atmosphere, $\Gamma_d$ ($\simeq -9.8\,\mathrm{K\,m^{-1}}$): $\partial\theta/\partial z > |\Gamma_d|$ is associated to an inversion, $0 < \partial\theta/\partial z < |\Gamma_d|$ to
a moderately stable atmosphere and $\partial\theta/\partial z < 0$ to an unstable atmosphere. The strength of the inversion is quantified by the
valley heat deficit (Whiteman et al. (1999a)):

$$\mathrm{VHD} = c_p \int\limits_{z_0}^{z_T} \rho(z)\left[\theta(z_T) - \theta(z)\right] dz \qquad \left[\mathrm{J\,m^2}\right] \tag{4}$$

where $c_p = 1005\ \mathrm{J\,K^{-1}kg^{-1}}$ is the specific heat capacity of air at constant pressure and $\rho$ is the air density; the lower and
upper bounds, $z_0$ and $z_T$, are the ground level (so $z_0 = 0$) and the elevation above ground level of the inversion top in the
Grenoble valleys. The quantity VHD is the amount of heat per unit area necessary to bring the temperature gradient of the
fluid column to the dry adiabatic lapse rate; in other words, it is the energy required to fully mix the air column and destroy the
inversion. From Largeron and Staquet (2016a), $z_T$ is set to 1500 m a.g.l. (this is the highest elevation of the inversion layers
observed in the Grenoble valleys during the winter of 2006-2007). The inversion height, denoted $H_{inv}$, is computed as the
highest elevation where $\partial\theta/\partial z > |\Gamma_d|$ below 1500 m a.g.l. (as in Largeron and Staquet (2016b) and Le Bouëdec (2021)).

   Figure 11 displays the temporal evolution of $\partial\theta/\partial z$, $H_{inv}$ and VHD, spatially averaged over the Grenoble basin, the Gré-
sivaudan valley, and the Voreppe valley. The inversion in Ep1988 is temporarily disturbed and destroyed between the night



of December 17 and the morning of December 18 (as previously observed in Figure 9), while the inversion height in Ep2043 decays down to a few hundred meters a.g.l. from December 10 and is disturbed on December 12. Therefore, we focus on the last three days (18-20 December) of Ep1988 and the first three days (8-10 December) of Ep2043 in the following analysis.

The $\partial\theta/\partial z$ values show that the atmosphere below the inversion height is more stable in Ep2043 than in Ep1988, with $\partial\theta/\partial z$

reaching values greater than $29.4\,\mathrm{K\,km^{-1}}$ not only in the surface layer during nighttime but also in a layer around 500 m a.g.l. (where the temperature abruptly changes in Figure 9). This difference in stratification between the two episodes is attested more precisely in Figure S7, which displays the vertical profiles of potential temperature averaged over the three selected days and nights of each episode.

Figure 11 also shows that the inversion height is lower in Ep2043 than in Ep1988, by about 100 m on average (see Table 3).

Consistent with these observations, the VHD is similar in both episodes, equal to about 11 $\mathrm{MJ\,m^{-2}}$. This value is of the same order of magnitude as those computed for real episodes (i.e. real-case simulations obtained with WRF forced by reanalysis): for the Grenoble valleys, values equal to 10–20 $\mathrm{MJ\,m^{-2}}$ have been found by Largeron and Staquet (2016b) for all PCAPs of the 2006-2007 winter and to 4–8 $\mathrm{MJ\,m^{-2}}$ by Le Bouëdec (2021) for four PCAPs in December 2013 and December 2016; for the Arve River valley (also in the French Alps), values equal to 8–12 $\mathrm{MJ\,m^{-2}}$ have been computed in Arduini et al. (2020)

during a PCAP in February 2015.

Note that the finding of similar VHD values for Ep1988 and Ep2043 is unchanged if the upper bound $z_T$ in definition (4) is set to the inversion height $H_{inv}$ instead of 1500 m. Values of VHD of the order of 7–8 $\mathrm{MJ\,m^{-2}}$ are then found for both episodes for the three valley branches.

A specific feature of a PCAP episode is that the inversion height $H_{inv}$ is insensitive to the diurnal cycle (see f.i. Largeron

and Staquet, 2016b). This is illustrated in Figure 11 for both episodes. During daytime, the inversion is eroded close to the surface by a shallow convective layer, of at most 50 m, while the atmosphere above remains stable. As a result, $H_{inv}$ correlates well with VHD, with correlation coefficients comprised between 0.67 and 0.85 (except for the Grésivaudan valley in Ep2043 where this coefficient is lower, equal to 0.53). This correlation can be approximately accounted for by noting that, when the potential temperature profile is linear, the valley heat deficit is proportional to the inversion height: $VHD = 0.5\rho c_p H_{inv}\Delta\theta$,

where $\Delta\theta$ is the potential temperature difference across the inversion layer (Whiteman et al., 1999b). Figure S8 shows that the linearity assumption can be made for the potential temperature averaged over episode Ep1988; for Ep2043, this assumption is not correct, which suggests that the proportionality between the VHD and $H_{inv}$ holds approximately beyond that assumption.

We note that $H_{inv}$ is higher in the Voreppe valley, lower in the Grenoble basin and even lower in the Grésivaudan valley both in 1988 and in 2043 (Table 3); the difference between Voreppe and Grésivaudan is about 100 m during Ep1988 and almost 200

m during Ep2043. The latter result is accounted for by the stronger valley wind in the Voreppe valley (see Figure S6) which promotes fluid mixing close to the ground, resulting in the raising of the inversion top (Chemel and Staquet, 2007).





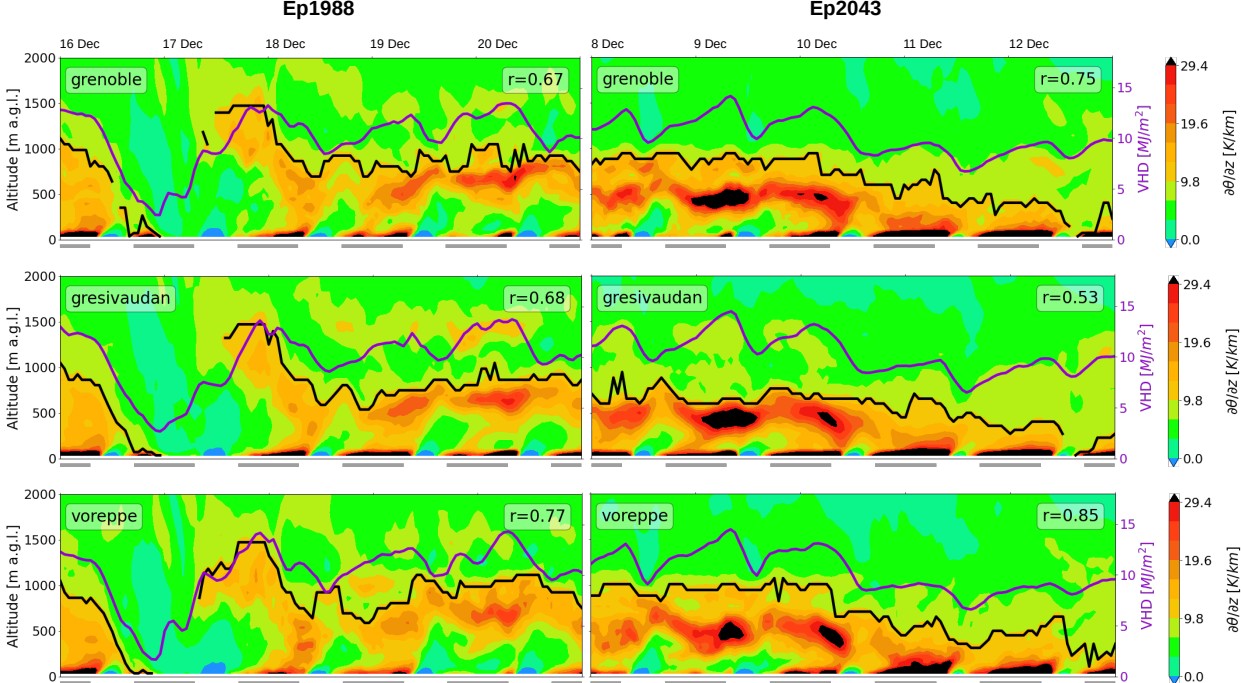

**Figure 11.** Vertical gradient of the potential temperature $\partial\theta/\partial z$ (color shading), inversion height (black line, left $y$-axis), and valley heat deficit VHD (purple line, right $y$-axis) for Ep1988 and Ep2043. These quantities are computed from the potential temperature, once temporally averaged (over one hour) and spatially averaged over the Grenoble basin, Grésivaudan valley, and Voreppe valley (see Figure 10). epThe coefficient $r$ is the correlation between $H_{inv}$ and VHD. The horizontal grey segments along the $x$-axis indicate the time between 17:00 UTC and 7:00 UTC.

| | $H_{inv}$ [m] | | VHD [MJ m$^{-2}$] | |
|---|---|---|---|---|
| | Ep1988 | Ep2043 | Ep1988 | Ep2043 |
| Grenoble basin | $877 \pm 122$ | $820 \pm 123$ | $11.0 \pm 1.4$ | $11.2 \pm 1.5$ |
| Grésivaudan valley | $824 \pm 153$ | $660 \pm 88$ | $11.1 \pm 1.4$ | $11.3 \pm 1.6$ |
| Voreppe valley | $922 \pm 161$ | $884 \pm 155$ | $11.3 \pm 1.3$ | $11.5 \pm 1.6$ |
| Overall | $874 \pm 49$ | $788 \pm 115$ | $11.2 \pm 0.2$ | $11.3 \pm 0.2$ |

**Table 3.** Temporal mean ($\pm$ standard deviation) of the inversion height and VHD computed over three days (18-20 December 1988 and 8-10 December 2043) and spatially averaged over the Grenoble basin, Grésivaudan valley, and Voreppe valley (see Figure 10).

# 6 Conclusions

We investigate the impact of climate change on persistent thermal inversions during wintertime in the Grenoble valley system during the 21st century. Persistent inversions are associated with persistent cold-air pools, called PCAPs. Our work relies on a



two-fold approach, statistical and deterministic. We first perform a statistical analysis of PCAP characteristics over the century from data of a regional climate model. Next we analyse the impact of climate change on two -carefully selected- PCAPs in the past and in the future using high resolution simulations.

The statistical analysis of the PCAP characteristics relies on outputs of the regional climate model MAR forced by the general circulation model MPI (MAR←MPI) from 1981 to 2100 for two different future scenarios, SSP2-4.5 and SSP5-8.5.

We propose a simple methodology to identify PCAPs from the MAR data, which we validate against observations. This methodology consists in computing wintertime daily-averaged vertical temperature gradients between the ground and the 925 hPa pressure level (i.e. about 600 m a.g.l., see subsection 2.3.1) assuming the temperature profile is linear during inversion episodes and using a threshold lower than 0 to detect the inversions (see relation (2) for more details). As shown in the literature (f.i. Caserini et al., 2017), using a threshold equal to 0 leads an underestimation of PCAP episodes.

The statistical analysis of the vertical temperature gradients over winter reveals a significant decreasing trend over the century both for SSP2-4.5 and SSP5-8.5. However, decay rates are very weak: 0.058 K km$^{-1}$ decade$^{-1}$ for SSP5-8.5 and four times lower for SSP2-4.5. Focusing on PCAP episodes, only for the former scenario is a statistically significant decreasing trend found, with a similar decay rate. Hence, the intensity of PCAP episodes is projected to be statistically stationary for SSP2-4.5 and to slightly decrease for SSP5-8.5. The tendency to a less stable atmosphere is likely due to the increasing near-surface

temperature. We also find that the PCAP mean duration remains basically unchanged for both scenarios but that the annual number of PCAPs is projected to decrease for SSP5-8.5, implying a smaller number of inversion days in winter for the latter scenario.

Criteria are then defined to carefully select comparable PCAPs in the past and in the future around 2050 (see subsections 2.3.2). This procedure leads to the selection of two PCAPs only, one in the past in December 1988 and one in the future in

December 2043. We next run the atmospheric numerical model WRF (using the model chain WRF←MAR←MPI) at high resolution (111 m) in order to simulate and compare the valley circulation and thermal structure of these two episodes.

The analysis of the two PCAP episodes, in the past and in the future, reveals that the temperature of the future episode is about 4 degrees warmer both close to the surface and in altitude than the one in the past. Also both episodes present similar atmospheric circulation and heat deficit across the valley depth but a different atmospheric stability and (therefore) inversion

height: the future episode is characterised by a stronger atmospheric stability and a lower inversion height. Hence intense PCAP can occur in the future.

To the best of our knowledge, this is the first study that investigates vertical temperature gradients and PCAP characteristics in an alpine mountain valley in a future warming climate. We anticipate that our results on the frequency of the PCAPs in the Grenoble valleys can be extended to other mountain valleys of the Alps and other mountain groups that are subject to the same

large-scale atmospheric circulation (f.i. Massif Central or Pyrénées in France). The occurrence and intensity of PCAPs are in fact strongly related to the weather pattern affecting the local climate. In the Iberian peninsula, for example, the most persistent and intense CAPs develop during anticyclonic conditions of the positive NAO phase (Rasilla et al., 2022). It must be also kept in mind that our results are based only on one GCM, the MPI model, and they could become not significant considering an ensemble of GCMs due to the inter-model variability; the EURO- and Med-CORDEX projects would contribute to extend the



analysis in this sense, by providing simulations coming from different model chains. On the other hand, the downscaling of MPI with MAR shows a very good performance compared to observations (section 3). Moreover, since the valley atmosphere will be less stable probably because of the increasing surface temperature and since all GCMs project such an increase until the end of the century, similar results for the vertical gradient trends may be obtained considering other model chains.

Overall, this study shows that the atmosphere in the Grenoble valleys tends to be as stable (for SSP2-4.5) or less stable (for SSP5-8.5) over the decades, although strong inversion episodes prone to poor air quality can still occur in the future. For the worst-case scenario, PCAPs will be less frequent and less intense. This less stable winter atmosphere could positively impact the air quality, as lower concentrations of particulate matter could be reached close to the ground during PCAPs.

*Data availability.* The MAR experiments over the Alpine domain are available on zenodo repositories for a limited number of levels and variables. MAR-MPI-ESM-HR, SSP2: https://doi.org/10.5281/zenodo.5834221; MAR-MPI-ESM-HR, SSP5: https://doi.org/10.5281/zenodo.5834376. More variables can be accessed by contacting the contact author. WRF outputs can be provided upon request to the contact author as well.

*Author contributions.* All authors contributed to designing the study. SB run WRF and JB run MAR. SB analysed the data and produced the figures, together with ELB. SB, JB, MM, HG, ELB, and CS discussed the results. SB and CS wrote the paper. All the authors provided assistance in finalizing the article.

*Competing interests.* The contact author has declared that neither they nor their co-authors have any competing interests.

*Acknowledgements.* This work was performed using HPC resources from GENCI-IDRIS (Grant 2021-A0080107161). The authors thank Michael Duda (NCAR) and Dave Gill (NCAR) for the discussion had at the beginning of this project and for their help to create the WPS intermediate file format. The authors thank also Milton Gomez (University of Lausanne) for having computed the vertical temperature gradients with the observations. SB thanks the Laboratory of Geophysical and Industrial Flows (LEGI) in Grenoble and the ADEME (PACC-MACS project) for having funded her scientific stay at the LEGI.



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
