# Peer review of "Impact of climate change on persistent cold-air pools in an alpine valley during the 21st century"

_EGUsphere, 2023_

## Author Comment (AC1)

**Authors' replies to Referee #1**

We thank Referee #1 for their helpful suggestions and comments. Below, we provide our replies. The line numbers in our replies refer to the revised version.

**General comments**

1. a. *The importance of surface warming as a driver of reduced stability during inversions is pointed out several times (line 336, 449, 472) and is important for the interpretation of the results. However, the detailed reason for such an increase of temperature during PCAP episodes is not explained clearly in the manuscript.*

   b. *The reference of Bailey et al. 2011 is provided at line 337 but I was not able to find relevant explanations there from a quick read.*

   c. *On the other hand, Fig. 9 shows an increase of around 4K for both surface and top of inversion (lines 369-374), and this might be indicative of warmer air at all levels due to advection, without obvious changes in stability.*

   d. *The authors need to point out the processes that would (eventually) enhance such a surface-based warming: for instance, are higher temperatures at the surface maybe due to changes in cloud cover, that modify the short-wave (daytime) or long-wave radiation balance?*

   a. We now explain the origin of this surface warming. It can be attributed to the fact that, in the future, specific humidity will be higher (i.e. air will be richer in water vapor due to higher temperatures) and, thus, will enhance the "local greenhouse effect", so that more infrared outgoing radiation is reflected back to the surface, increasing air temperature close to the surface. We explain this at the end of subsection 4.1 (L 347-349); we also include a new Figure (Figure S6) in the supplement showing the statistically significant increasing trend of specific humidity close to the surface. Reference to Philipona (2013) is also added, who pointed the importance of specific humidity in surface warming.

   We also computed the temperature trends at 925 hPa and 850 hPa (new Figures S5 and S6 and L 339-347). These trends are significant and slightly lower than that of the temperature at 2 m, confirming that the negative trend of the PCAP temperature gradient is due to surface warming.

   b. We removed the reference to Bailey et al. 2011.

   c. The temperature difference between the two episodes is best seen by displaying the vertical profiles of the temperature. These profiles have been added in the new Figure S9 (first row), see also L 384-387. They show that the future episode displays a marked inversion, which is stronger than for the episode in the past.

   d. We do not think that incoming and outgoing radiation is influenced by different cloud cover during the two episodes for the following reasons: the two episodes are characterised by anticyclones over Grenoble, so it is reasonable to suppose that there is no/scarce cloud cover in both episodes; moreover, if there were cloudy days, we would not see the temperature diurnal cycle near the surface that we can instead observe in Fig. 9.

2. *According to the performed simulations, the vertical stability seems to increase for stagnation events in a warmer climate, especially for the elevated thermal inversions (Fig. 11). This hints to the presence of warmer air above the inversion in the 2043 event, likely resulting from non-local processes such as advection. Together with a reduction in the height of the BL of ~100m, the model simulations would thus indicate a strengthening of thermal inversions, related mostly to the warming at upper-levels. This would be in contrast with the current interpretation of the results, which indicate an average reduction in inversion strength during the next century driven by surface-based warming. Can the authors please reconcile these contrasting results? Are there*

*reasons to expect extremes to follow a different trend than the mean?*

We would like to stress that this work consists in two types of analyses (statistical and deterministic, as written at L 83-86 of the Introduction) which allow to infer conclusions that are not in contrast (as now specifically added at L 387-390 and 471-473) but of different type. Therefore, we cannot assert a general sentence like the one above "*the vertical stability seems to increase for stagnation events in a warmer climate, especially for the elevated thermal inversions*" because this consideration exclusively refers to the comparison between Ep1988 and Ep2043, which are only two episodes among hundreds of possible episodes along the century. While we can study the impact of climate change on PCAPs with the first type of analysis, based on long-term trends, we can investigate the vertical structure of two PCAPs, one in the past and one in the future, with the second type of analysis. The results obtained for Ep2043 do not reflect the statistical trend computed over 120 years, but there is no contradiction because we analyse only two episodes. We now make this important point clear at the lines indicated above.

We also changed the titles of sections 5.1 (L 376) and 5.2 (L 405) to clarify that the analysis of section 5 is conducted for "the two PCAP episodes".

3. *Given the fact that VHD does not appear to change substantially between the two events, can we conclude that what changes for PCAP events in a warmer climate is not the rate of cooling but rather the initial temperature at which cooling starts?*

   As written in the previous reply, we cannot infer any general conclusion for "PCAP behavior in a warmer climate" from the deterministic analysis of two only episodes.

   Regarding the VHD: this quantity is a bulk measure of stability and should therefore reflect the fact that the future episode is more stable than the past one. The value of the VHD actually depends upon the upper bound of the integral defining it. It is therefore important this upper bound to coincide with the height of the cold-air pool. In the submitted version, this upper bound was well above the top of the cold-air pool and, as a result, the values of the VHD were similar. We now adjusted this upper bound so that it coincides with the height of the cold-air pool (see L 415-416) and found that VHD is larger for the future episode than for the past one, as expected (see new values in Table 3). This result is also consistent with the fact that, when the potential temperature profile is linear, the VHD is related to height of the cold-air pool by the relation $VHD = 0.5\rho c_p H_{inv}\Delta\theta$ where $\Delta\theta$ is the temperature difference across the inversion (see L 441-443).

4. *It is not yet clear how the projected trends in inversion height and strength will reflect themselves in air quality, and some apparently contradictory statements are found in the manuscript. For instance, at lines 474-476 is written "the less stable winter atmosphere could positively impact the air quality", but previous results (summarized at line 460) indicate that future episodes will feature "a lower inversion height", which would worsen air quality. The contradiction is mostly between the effects of inversion strength and inversion height, but I would think that the latter is more important than the former, provided that a sufficiently strong inversion does not "break" during daytime.*

   As stressed in the reply to point 2, we must differentiate the conclusions derived from the two types of analysis. From the statistical analysis, we find a negative trend of inversion stability over the century that suggests that the valley atmosphere will be less stable. However the decay rate of the temperature gradient inside the cold-air pool is so small (0.057 K/km per decade for scenario SSP5-8.5, this rate being non significant for scenario SSP2-4.5) that we cannot conclude that air quality will improve. This is what we write in the Conclusion (L 493-495).

   From the deterministic analysis, which compares the two selected episodes, we find that the

[Figure]

Figure 1: Weather types: WT1 = Scandinavian blocking, WT2 = Atlantic Ridge, WT3 = Positive NAO, WT4 = Negative NAO.

cold-air pool of Ep2043 has a strong stability so, during this type of episode, air quality will worsen. This is what we add on L 496-497.

**Technical/Typos/Etc...**

- *Line 226: which value has been "rounded" and how?*
  The 30-year mean is -3.136 K/km; since it is close to -3 K/km, we rounded it to the nearest integer (as done in other works cited in the manuscript), i.e. to -3 K/km. We do not think it is relevant to add the value of -3.136 K/km in the manuscript.

- *Line 245-246: the definition based on 4 weather regimes confounds together Scandinavian blocking, European blocking and sometimes even Greenland blocking, that other authors would indicate as responsible of stagnation events over central Europe (e.g., 10.1088/1748-9326/ab38d3). Please provide reference(s), or indicate that this statement only refers to the k=4 choice for weather regimes.*
  We are not sure in which sense the 4 regimes are confounded. In the reference cited by the Referee the four weather regimes are computed in the same way as done in this paper (apart from the fact that the four centroids are based on the reanalysis instead of on the GCM) and the patterns of the four weather regimes are totally in agreement with those in this work (see Figure 1 of this document). Only the so-called "Scandinavian blocking" is responsible for permanent anticyclonic conditions over France (like in Fig. S2) and, therefore, we focused on this weather regime; we specified this better in the revised version (L 248-257).

- *Line 269: at which level is the wind estimated?*
  The wind is estimated at 500 hPa (this is written in the caption of Fig. S2). We specified the level by writing: "The winds at 500 hPa over South-East France are..." in the revised version.

- *Line 381: "wind intrusion" is not commonly used and might be misleading, please use other formulations depending on the meteorological object associated with that wind maximum (e.g., is it a jet streak, maybe related to a small upper-level trough?).*
  We changed the expression into "cold-air subsidence". We must admit that we did not dig into the meteorological reason for this cold-air subsidence as it is not of interest for the analysis.

- *Lines 425-427: this sentence is not clear, because if the assumption is "not correct", how can the assumption "approximately hold"? Please specify why the linear assumption is not anymore a good one for Ep2043, and whether the linear extrapolation to compute VHD is still useful.*
  This sentence has been removed.

- *Line 447: scenario corresponds to a statistically.*
  Thanks. The sentence was corrected (i.e. "only the former scenario shows a statistically significant decreasing trend", L 458).

- *Line 468: in which sense "become not significant"? This terms should be used only in the context of statistical tests.*
  We agree with the Referee. The sentence was rephrased explaining that an ensemble of GCMs would take into account the inter-model variability and allow for the estimate of the model uncertainty (L 485-488).

- *Table 2: add asterisks or denote otherwise which changes are significant.*
  Given the high variability within 30-year periods (visible in the temporal series in Fig. 8), there a no statistically significant results in Table 2 in the sense that future values (for the two 30-year periods and for the two scenarios) are always within the interval [(mean around 2000) ± (standard deviation)].

- *Fig. 4, 5, would profit of small titles indicating which quantity is being looked at (e.g., "PCAP duration", "PCAP stability", etc. . . ).*
  Plot titles have been added in Figures 4 and 5.

- *Fig. 9, 11: mark the three days period chosen for averaging in both figures (lines 402-403).*
  We marked the three days with a blue rectangle in both figures of the revised version (which are now Figures 9 and 10 as the former Figure 10 has been moved to the supplementary material).

- *Fig. 11: epThe*
  Thanks for noticing this typo.

---

## Author Comment (AC2)

**Authors' replies to Referee #2**

We thank Referee #2 for their helpful suggestions and comments. Below, we provide our replies. The line numbers in our replies refer to the revised version.

We make it clear that the paper contains two independent analyses, one is the statistical analysis over the century of the vertical temperature gradient inside the valley from MAR data, the other one aims at analysing the vertical structure of the temperature and horizontal velocity fields inside the valley from two episodes, one in the past and one in the future. These analyses are not connected and, therefore, they cannot be in contrast. The fact that the two episodes (selected from common criteria so as to perform a meaningful comparison) do not reflect the statistical trend is not surprising since only two episodes are selected.

**Minor points:**

1. *Lines 23-38: The first paragraph of the introduction could be improved. It reads a bit bumpy. Moreover, I think that you should describe the implications of PCAPs on society in more detail and introduce the relevance of your study for society, i.e. why it is important to study this phenomenon.*

   We improved and updated the Introduction following the Referee's suggestions (L 35-41).

2. *Line 36: "The qualificative ground-based,..." - can you please rewrite the sentence or put the terms ground-based, surface-base etc in italics? This would be easier to understand.*

   We used the italic style for these terms (L 50-51).

3. *Line 117: "(CMIP6, Eyring et al. (2016))" - should be: (CMIP6, Eyring et al., 2016).*

   We changed it in the revised version of manuscript (L 123).

4. *Line 189: Is "Noah Land surface model" correct? Moreover, I cannot find the reference Chen and Dudhia (2001) in your reference list!*

   We confirm that the land surface model used in WRF simulations is the Noah LSM (this is set in the namelist with *sf_surface_physics = 2*; all option of the namelist are visible in this link).
   Thanks for noticing that the reference is missing; we added it.

5. *Figure 6: Can you try to plot a violin plot or a box plot. This could improve your plots. It is not easy to grab information from these pure scattered points.*

   Plots in Fig. 6 are not scatter plots but temporal series of daily winter temperature gradient, plotted with a dot mark for each day (instead of using a line). The objective of these plots is showing the "raw" data and the trend over the century. Moreover, they are useful for Fig. 7, as they help visualizing the windows of variable length considered in this figure. A box plot (which "collects" data) would obscure the key point of the Fig. 6, which is the decaying trend, and, therefore, would be not appropriate.

6. *Figure 7: Can you please again explain in more detail: why do you see the significance of trends for larger periods, but not for the smaller ones?*

   The color of each "pixel" in Fig. 7 indicates the slope of the trend computed within a window of length equal to the value on the y-axis, centered in the year equal to the value on the

x-axis, moving along the temporal series of Fig. 6. For each window, the statistical significance has been computed using the p-value of the null hypothesis test. When the result is statistically significant at 95%, i.e. p-value $< 0.05$, we marked the pixel with a black cross. For the worst-case scenario, we can see that all pixels for window length $= 80$ up to the vertex of the triangle are marked; this is the reason of our sentence "the trends are [...] and always statistically significant with windows longer than 70 years". Only longer periods present statistically significant trends because of the high variability of the temporal series (visible in Fig. 6).

7. *Figure 8: Do your trends stay significant if you choose different periods, for example 2020-2080? The variability seems so high, that I wonder what will happen, if you change periods (for example by rolling over 20 years).*

   Like in Fig. 7, where we see the variability of the statistical significance, also in this case the statistical significance of the yearly PCAPs characteristics is variable, as the Referee correctly imagined. We made some tests for different periods with windows of 100, 80 and 60 years (while the "window" considered in Fig. 7 corresponds to the entire time series, i.e. 120 years):
   - 2000-2100 (window of 100 years): we obtain a statistically significant negative trend only for the yearly mean of PCAP dT/dz for SSP5-8.5;
   - 2000-2080 (window of 80 years): no significant trends;
   - 2020-2100 (window of 80 years): we obtain a statistically significant negative trend only for the yearly mean of PCAP dT/dz for SSP5-8.5;
   - 2020-2080 (window of 60 years): this time, we obtain a statistically significant negative trend for the yearly number of PCAP episodes for SSP5-8.5.

8. *Figure 9: It is really difficult to see the warm temperature values on the left figure. I understand, that your goal is to show that it is much warmer in the future. However, it would be great if you can additionally plot the figure by normalized values (for example between minimum (=0) and maximum (=1)? This figure could also be added to the supplementary material if you do not want to add another figure to your main paper.*

   We thank the Referee for their suggestion. We produced Figure ?? displaying the temperature rescaled between minimum and maximum values over the three areas of the Grenoble valley, for each episode. This figure clearly shows similarities and differences between the two episodes. However, following the Reviewer's comment, we think it is more useful to display the temperature profile for each episode (new Figure S9 upper row). For this reason we did not include Figure 1 in the paper.

9. *Line 391, equation (4) and line 425-427: Can you please explain why you do not expect much difference if you set z_t to 1500 m or to the inversion height (if I understood it correctly?). Maybe you can give some simple, idealized examples (for different vertical profiles) to explain your assumption?*

   The VHD is a bulk measure of stability and should therefore reflect the fact that the future episode has a stronger stability than the past one. The value of the VHD actually depends upon the upper bound of the integral defining it. It is therefore important this upper bound to coincide with the height of the cold-air pool. In the submitted version, this upper bound was well above the top of the cold-air pool and, as a result, the values of the VHD were similar for the two episodes. We now adjusted this upper bound so that it coincides with the height of the cold-air pool (see L 415-416). We found that VHD is larger for the future episode than for the past one, as expected (see new values in Table 3 and L 435-436). There is no contradiction with our statement in the submitted version where we wrote that "values of VHD are of the order of 7-8 MJ/m$^2$": in overall indeed VHD values for Ep1988 are equal to 7.3 MJ/m$^2$ and for Ep2043 equal to 8.2 MJ/m$^2$, namely VHD is all the higher the stability is stronger. This is also consistent with

[Figure]

Figure 1: Normalized temperature field computed as $(T - T_{min})/(T_{max} - T_{min})$ where $T_{min}$ and $T_{max}$ are the minimum and maximum air temperature, respectively, among the three valleys (each episode has one $T_{min}$ and one $T_{max}$).

the fact that, when the potential temperature profile is linear, the VHD is related to height of the cold-air pool by the relation $VHD = 0.5\rho c_p H_{inv}\Delta\theta$ where $\Delta\theta$ is the temperature difference across the inversion (see L 441-443).

10. *Line 409 (see also Table 3): You say that the inversion height is on average 100m lower in the future, however, this is not significant! Please clarify!*

    Yes, we agree, we now write that the two episodes have similar inversion heights (L 431-433).

11. *Line 410: "Consistent with these observations ..." - Why is this congruent? In the preceding sentence, you noted a lower inversion height, while here, you mention a similar Vertical Heat Diffusivity (VHD). Can you please provide clarification?*

    As noted above, we rewrote this part of the section dedicated to the analysis of VHD and the PCAP height.

12. *Lines 425-427: I do not understand the sentence starting with "Figure S8 [..]". Please explain.*

    Following also the comment by Referee #1, we removed these two sentences.

13. *Line 460: I think that you cannot draw the conclusion that the inversion height in the future is generally lower. Please clarify that you just looked at two example periods. Since the variability between episodes is large, in my opinion it is not possible to draw such conclusions.*

    We clarified the behavior of inversion height on L 428-433.

14. *Line 486: There is a word missing in the sentence.*

    Thanks a lot for noticing this.

---

## Author Response (AR2)

**Authors' replies to the Referee**

We thank a lot the Referee for their comments and further suggestions. Below, we provide our replies.

**Minor comments**

1. *The authors noticed the existence of a significant specific humidity trend in their simulation, and exploited this observation to explain the observed surface temperature increase and associated stability reduction during PCAP events. Even the proposed mechanism is physically plausible, the connection between the humidity and the temperature trend is not shown explicitly. Would it be possible to check whether trends in incoming long-wave radiation match the observed specific humidity trend, in general and/or during PCAP events? Without this type of analysis, the statement at line 12 in the abstract would need to be toned down, for instance by replacing of "is due to" with a more hypothetical formulation (e.g., "might be due") as done in the Conclusions.*

   We agree with the Referee that the sentence in the abstract is too "strong". We rephrased the sentence as follows: "This decay is explained by the fact that air temperature over the century increases more at 2 m above the valley bottom than in the free air at mid-altitudes in the valley; this might be due to the increase of specific humidity near the ground."
   (Unfortunately, the outputs of MAR←MPI for radiation, longwave and shortwave radiation, were not saved.)

2. *Could the authors explain how trends are computed, and in particular how serial correlation is taken into account when assessing trend significance? If trends are computed on all points displayed in Fig. 6 (daily means), there is a risk that positive auto-correlation might lead to spurious trends and detection of significance (e.g., https://doi.org/10.1016/j.earscirev.2018.12.005).*

   The trends are computed as linear regressions based on the daily means of winter (NDJFM) periods (i.e. all the points displayed in Fig. 6). The python function "scipy.stats.linregress" has been used for the computation. We added in the manuscript that the trends are computed as linear regressions in the captions of Fig. 6 and Fig. 8; thanks for noticing this missing information.

   We also thank the Referee for the interesting reference of Mudelsee 2019. We would like to explain that we decided not to compute the linear regression after the application of a running mean (one of the methods suggested in the reference) for the following reason: the temporal series we consider do not evolve on a continuous time axis as summer days are missing, therefore, we would have averaged values of March with values of November in the same window. We rather preferred to compute the trend on the original time series as it contains more than 18000 points which should allow for a robust computation of the trend. Finally, the autocorrelation mentioned in Mudelsee 2019 is interrupted once per year, in our case, due to the selection of winter periods only.

   Nevertheless, we investigated some aspects mentioned by Mudelsee 2019 and we performed the following computations. We computed the correlation between the residuals of two daily temporal series (i.e. $(\Delta T/\Delta z)_{MAR}$, which we will call $X(i)$, with $i$=days) shifted by $\Delta i$. The residuals are computed as $R(i) = X(i) - (at(i) + b)$, where $a$ and $b$ are the coefficients of the linear regression. We obtained that, if $\Delta i$ is equal to one day ($\Delta i = 1$), the Pearson correlation between $R(i)$ and $R(i+1)$ is about 0.53; such a (not low) correlation is expected since $\Delta i$ is equal to one day only (we remind that Mudelsee 2019 considers annual means in the temporal series, instead of days). If we increase the shift by one more day, namely $\Delta i = 2$, the correlation becomes very low: the Pearson correlation between $R(i)$ and $R(i+2)$ is about 0.27 and the scatter plot is a cloud.

Finally, we also computed the correlation of the residuals of $X(i)$ and $X(i+1)$ where $X(i)$ is the annual (i.e. winter) mean of $(\Delta T/\Delta z)_{MAR}$ (i.e. using annual time series, like in the example of Mudelsee 2019, Figs. 2 and 3). This time, the correlation is really close to zero, showing no autocorrelation between years. We therefore believe that the use of the linear regression method for the trend computation in our paper is justified.

3. *Line 402: "cold air intrusion" would be a clearer terminology in this context, as "subsidence" is usually associated with descent and adiabatic warming. Besides being confusing, no analysis of vertical motion is performed in the paper.*

We thank the Referee for the suggestion. We used the expression "cold air intrusion" instead of "cold-air subsidence". (The analysis of air vertical motion, especially above the inversion top, was not the focus of this study.)

---

## Author Response (AR3)

**List of changes made in the final version of the manuscript**

Dear Editor,

thank you very much for your positive comment on our work. We removed part of L10 (Abstract), as suggested.

Yours faithfully,

Sara Bacer